# Lifelines COVID-19 cohort: investigating COVID-19 infection and its health and societal impacts in a Dutch population-based cohort

Katherine Mc Intyre [1], Pauline Lanting [1], Patrick Deelen [1,2]
Henry H Wiersma [1], Judith M Vonk [3], Anil P S Ori [1,4]
Soesma A Jankipersadsing [1], Robert Warmerdam [1], Irene van Blokland [1,5]
Floranne Boulogne [1], Marjolein X L Dijkema,[1] Johanna C Herkert [1]
Annique Claringbould [1], Olivier Bakker [1], Esteban A Lopera Maya [1]
Ute Bültmann [6], Alexandra Zhernakova [1], Sijmen A Reijneveld [6]
Elianne Zijlstra [6], Morris A Swertz [1], Sandra Brouwer,[6] Raun van Ooijen [6]
Viola Angelini [7], Louise H Dekker,[7,8] Anna Sijtsma,[9] Sicco A Scherjon [10]
Cisca Wijmenga [1,11], Jackie A M Dekens,[1,12] Jochen Mierau [7,13]
H Marike Boezen [3], Lude Franke [1]

► Prepublication history and supplemental material for this paper is available online. To view these files, please visit the journal online (http://dx.doi.org/10.1136/bmjopen-2020-044474).

PL, PD, HHW, JMV, APSO and SAJ contributed equally.

JAMD, JM, HMB and LF are joint senior authors.

For numbered affiliations see end of article.

**Correspondence to**
Professor Lude Franke;
l.h.franke@umcg.nl

## ABSTRACT

**Purpose** The Lifelines COVID-19 cohort was set up to assess the psychological and societal impacts of the COVID-19 pandemic and investigate potential risk factors for COVID-19 within the Lifelines prospective population cohort.

**Participants** Participants were recruited from the 140 000 eligible participants of Lifelines and the Lifelines NEXT birth cohort, who are all residents of the three northern provinces of the Netherlands. Participants filled out detailed questionnaires about their physical and mental health and experiences on a weekly basis starting in late March 2020, and the cohort consists of everyone who filled in at least one questionnaire in the first 8 weeks of the project.

**Findings to date** >71 000 unique participants responded to the questionnaires at least once during the first 8 weeks, with >22 000 participants responding to seven questionnaires. Compiled questionnaire results are continuously updated and shared with the public through the Corona Barometer website. Early results included a clear signal that younger people living alone were experiencing greater levels of loneliness due to lockdown, and subsequent results showed the easing of anxiety as lockdown was eased in June 2020.

**Future plans** Questionnaires were sent on a (bi)weekly basis starting in March 2020 and on a monthly basis starting July 2020, with plans for new questionnaire rounds to continue through 2020 and early 2021. Questionnaire frequency can be increased again for subsequent waves of infections. Cohort data will be used to address how the COVID-19 pandemic developed in the northern provinces of the Netherlands, which environmental and genetic risk factors predict disease susceptibility and severity and the psychological and societal impacts of the crisis. Cohort data are linked to the extensive health, lifestyle and sociodemographic data held for these participants

## Strengths and limitations of this study

► The Lifelines COVID-19 cohort collects data about factors relevant to the impact of the COVID-19 pandemic for >70 000 individuals living in the Northern Netherlands. Participants in the cohort are also participants in the Lifelines prospective population cohort and the Lifelines NEXT mother–baby cohort, which means there is already a rich data background for all participants, and cohort data can be linked to data held in other national databases.

► The cohort questionnaire programme began during the height of the first wave of infections in the Netherlands and continued through late 2020 and early 2021. The questionnaires were designed by a multidisciplinary group of researchers to explore factors that may impact COVID-19 susceptibility and severity and the social, mental and economic impacts of the pandemic.

► The compiled questionnaire data have been continuously shared with the participants and the public through an interactive website, and new questionnaire modules have been added to address questions raised by participants and researchers.

► The region covered by the Lifelines COVID-19 cohort has so far experienced very low numbers of COVID-19 cases, as compared with the rest of the Netherlands. This reduces the power to detect COVID-19 related risk factors but makes the cohort an interesting resource for examining the broader mental health impacts of the governmental measures to slow infection rates and the associated economic slowdown.

by Lifelines, a 30-year project that started in 2006, and to data about participants held in national databases.

## INTRODUCTION

COVID-19 has now impacted the lives and health of billions of people around the world. Due to the initial absence of a vaccine, lack of effective antiviral medication and limited understanding of the SARS-CoV-2, most governments have tried to slow the growth rate of infections through public health measures including tracking and testing, shutting down of public life, social distancing policies and stay-at-home orders. These measures have had a huge impact on public health and well-being, the economy (including employment and working conditions) and daily life. The effects of the COVID-19 pandemic will therefore be multiple: there will be the impact of the infection itself and the broader societal and health impacts.

To identify genetic and environmental risk factors for COVID-19 and address the medical, social and psychological impacts of the pandemic, a multidisciplinary group of researchers rapidly developed and implemented an extensive COVID-19 questionnaire, leading to the development of the Lifelines COVID-19 cohort. The questionnaire collects data about COVID-19 related symptoms, current health issues and societal impacts from participants recruited from the Lifelines population cohort[1] and the Lifelines NEXT (LLNEXT) birth cohort,[2] which are both monitoring the health of the northern Dutch population (provinces of Drenthe, Groningen and Friesland). Via a (bi)weekly questionnaire, the project gathers information about COVID-19 symptoms, associated comorbidities and environmental factors, changes in work and employment, COVID-19 related worries, loneliness and the mental health and societal impacts of the pandemic. In addition, all participating parents are asked about their children's well-being, and LLNEXT parents received detailed questions about COVID-19 related symptoms expressed by their children. Additional questionnaire modules have been included as the project progressed.

The data collected by the questionnaires is being used to address four aspects of the outbreak: (1) how the COVID-19 pandemic developed in the three northern provinces of the Netherlands, (2) which environmental risk factors predict disease susceptibility and severity, (3) which genetic risk factors predict disease susceptibility and severity and (4) the psychological and societal impacts of the crisis.

### The initial COVID-19 outbreak in the Netherlands and the northern provinces

The first official COVID-19 cases in the Netherlands were registered on 27 February 2020 (see timeline figure 1A).[3] By 24 March 2020, the number of cases diagnosed per day had risen to 1126 (figure 1B). The rapid rise in case numbers led the Dutch government to shut down primary and secondary schools, bars and restaurants,

sporting facilities and other public spaces on 15 March 2020, followed by a more extensive shutdown of public life in the weeks that followed (see figure 1A for major events). However, the three northern Dutch provinces did not follow national trends. COVID-19 appeared later here and did not reach the same incidence of cases or infection rates. While the three northern provinces account for 10% of the Dutch population, they only had 2%–3% of cases, hospitalisations and deaths in the period from 27 February to 9 June 2020 (see figure 1C; online supplemental table 1). Multiple factors may explain why the outbreak was different in the north.[4 5] Drenthe, Groningen and Friesland together are the least populated region of the Netherlands and contain the fewest urban centres. The early arrival and spread of infection in the southern Dutch province of North Brabant seems to have originated in travel to and from Northern Italy during the school holidays from 22 February–1 March 2020, with the spread of the infection in the southern Dutch region then further facilitated by personal contact during regional carnival celebrations. In contrast, school holidays fell earlier for the northern provinces (15–22 February 2020), which suggests that northerners who travelled to Italy during the vacation had returned before the major expansion of the outbreak in Northern Italy.[6] Nor is carnival widely or generally celebrated in the northern provinces. Other factors may also have played a role, for example, use of a stricter COVID-19 testing regime and wider availability of tests in the region.[5] The later arrival of COVID-19 to the north meant that the national steps taken to bring down the infection rate were in place before the outbreak had really taken hold in the Lifelines region.

## COHORT DESCRIPTION
### Participant recruitment

Participants of the Lifelines COVID-19 cohort are recruited from the Lifelines and LLNEXT cohorts. Lifelines is a prospective population cohort following ~167 000 people in the three northern provinces of the Netherlands for 30 years[1] (see figure 2 for Lifelines region). Established in 2006, Lifelines collects detailed information about its participants via extensive questionnaires and medical examinations, and the cohort has been shown to be representative of the Northern Dutch population.[7] By design, Lifelines recruited multiple participants within families to produce a multigenerational cohort that could map individual and community health across life course.[1] Since 2016, LLNEXT has been recruiting an additional generation through inclusion of mother–baby pairs, with partners also invited to participate to generate parent–baby trios.[2]

To recruit participants for the Lifelines COVID-19 cohort, Lifelines and LLNEXT invited their participants digitally to fill out the questionnaires. In each questionnaire round (Q1–Q7 in figures 1D and 3), all Lifelines participants over the age of 18 years with a known email

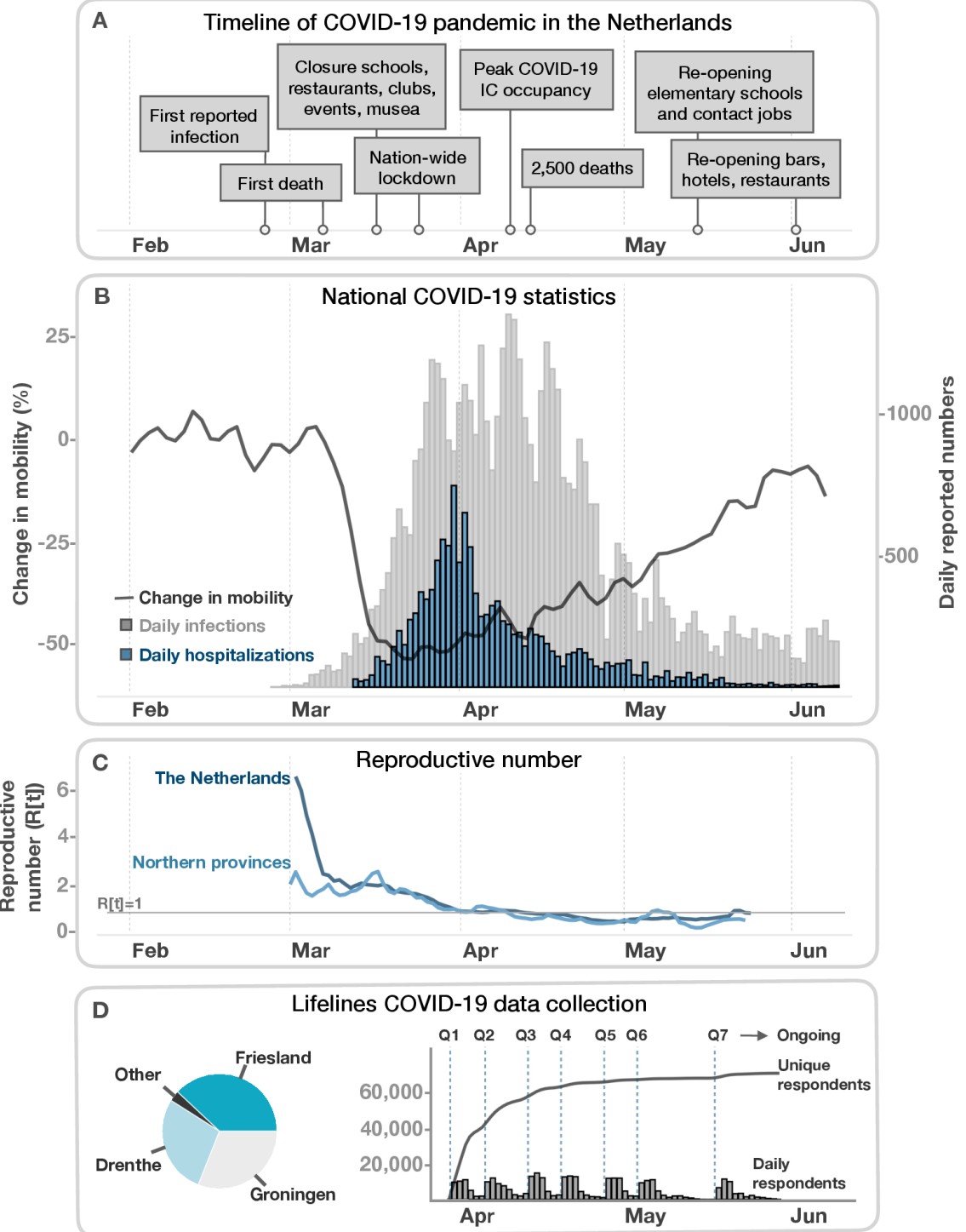

**Figure 1** Timeline of the COVID-19 pandemic in the Netherlands and Lifelines data collection. (A) Important events of the pandemic in the Netherlands from February to June 2020. (B) Daily reported positive infections (grey) and hospitalisations (blue) visualised alongside the change in mobility (black) in the Netherlands. Mobility is quantified using Apple Maps Request data (https://www.apple.com/COVID-19/mobility) with the change over time normalised to 1 February 2020. Change in mobility indicates the percentage change in overall requested driving directions by users of Apple Maps. COVID-19 daily infections and hospitalisations are derived from the CoronaWatchNL github account (https://github.com/J535D165/CoronaWatchNL) and are based on reported numbers from the Rijksinstituut voor Volksgezondheid en Milieu (RIVM). (C) The reproductive number in the Netherlands and the three northern provinces over time. The R(t) is calculated based on incident cases (new positive PCR tests) including healthcare workers and cases appertaining to local outbreaks. National and regional R(t) values in the early phase of the pandemic are not directly comparable, since testing among healthcare workers was more widely adopted early on in the northern provinces. (D) Overview of the Lifelines COVID-19 data collections. The pie chart on the left shows the proportion of participants for each province. The first weekly COVID-19 questionnaire (Q1) was sent out on 30 March 2020. Based on Q1–7, 71 800 unique respondents have filled out at least one questionnaire. From Q7, assessments are biweekly. IC, intensive care.

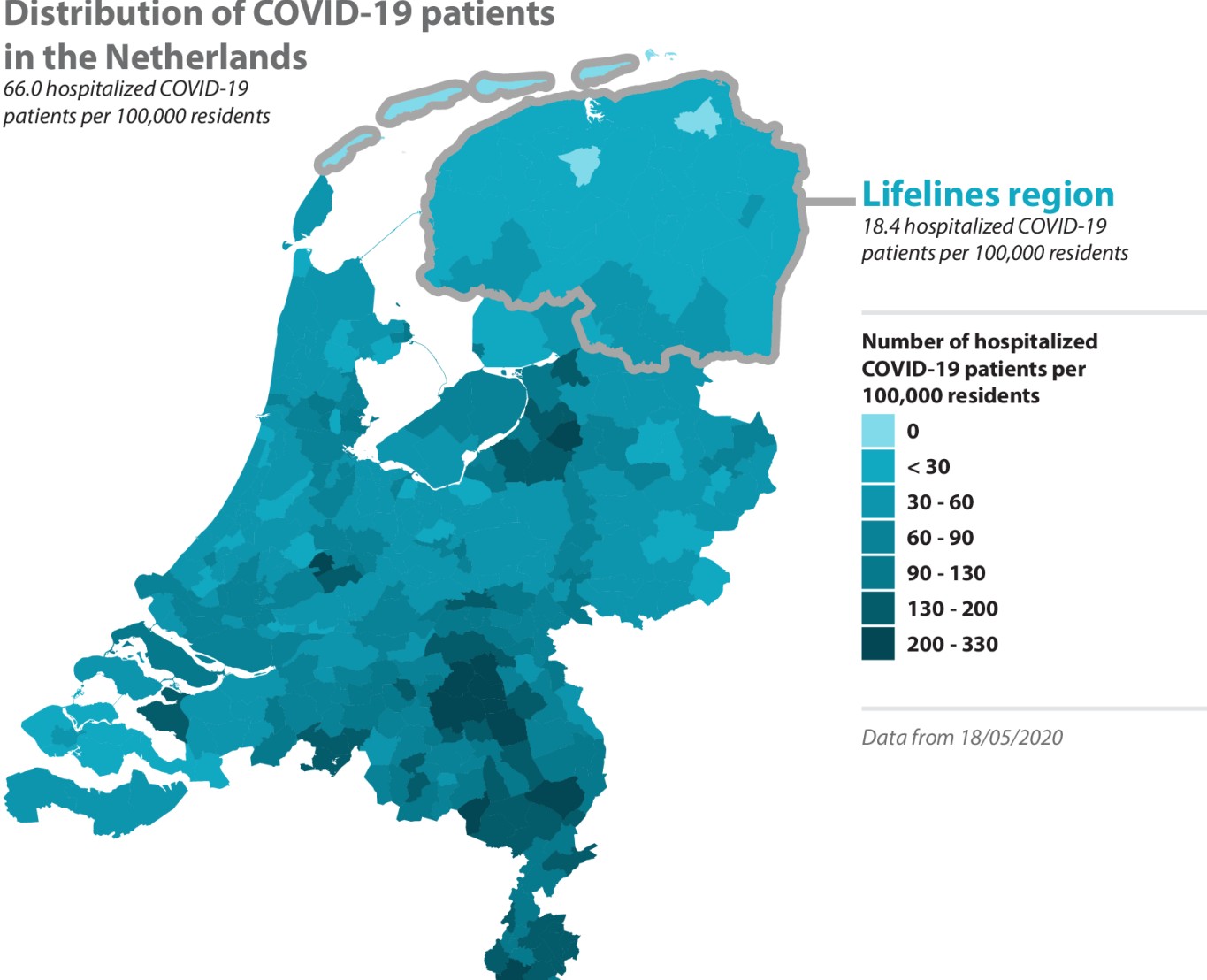

**Distribution of COVID-19 patients in the Netherlands**

*66.0 hospitalized COVID-19 patients per 100,000 residents*

**Lifelines region**

*18.4 hospitalized COVID-19 patients per 100,000 residents*

**Number of hospitalized COVID-19 patients per 100,000 residents**

- 0
- < 30
- 30 - 60
- 60 - 90
- 90 - 130
- 130 - 200
- 200 - 330

*Data from 18/05/2020*

**Figure 2** Distribution of hospitalisation across Dutch municipalities. The number of hospitalisations per municipality, as reported by the Rijksinstituut voor Volksgezondheid en Milieu (RIVM), were integrated with a geographical map of the Netherlands. For each municipality, the cumulative number of COVID-19 hospitalisations per 100 000 residents is shown. The lifelines region is outlined in grey. These data were downloaded on 18 May 2020.

address received a link to the digital COVID-19 questionnaire (see inclusion flow chart in figure 3). Digital invitations were valid for 3 weeks, and the date on which the questionnaire was completed was registered. Since LLNEXT is an ongoing project in which new participants are still being included, the number of LLNEXT participants invited to participate in the COVID-19 cohort increased with each new questionnaire round. In each round, invited participants chose if they wanted to fill in the questionnaire, and the cohort population consists of all those who filled in at least one questionnaire in the first seven questionnaire rounds (figures 1D and 3).

On 30 March 2020, all Lifelines and LLNEXT participants were invited to participate in the first COVID-19 questionnaire round (Q1 in figures 1D and 3), with new invitations to participate sent out weekly. Starting 27 April 2020, an additional questionnaire about children's

health and symptoms was sent to the participants of the LLNEXT (>300 participants). Programme questionnaires were sent out weekly through the week of 18 May 2020, then at biweekly intervals until July 2020, when the questionnaires became monthly. The project has continued through 2020 and early 2021, with the option to increase questionnaire frequency when the local caseload begins to increase rapidly. While the timeline of the questionnaire programme will be decided by the outbreak, the cohort itself will continue to exist as part of Lifelines, which will allow participants to be monitored for the long-term health impacts of the pandemic through the length of the Lifelines programme.

**Questionnaire contents**

The Lifelines COVID-19 questionnaire includes question modules about sociodemographic parameters, chronic

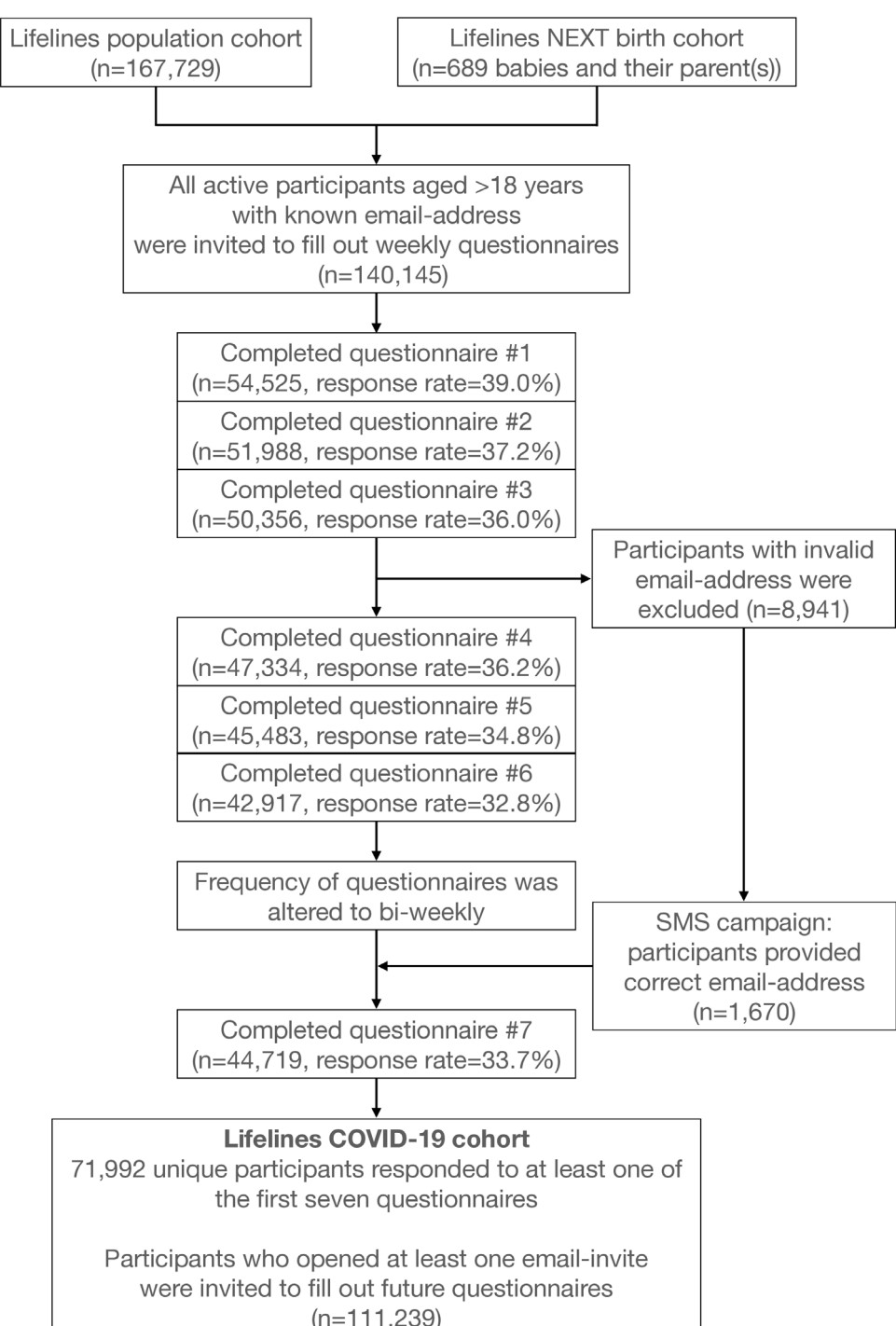

**Figure 3** Study inclusion flow chart.

diseases, COVID-19 infection, general health and symptoms, medication use, the mental health/well-being of participants and of children and young adults in their family, COVID-19 related well-being, social life, social relations and lifestyle (see table 1 for modules and questions and https://www.lifelines.nl/researcher/data-and-biobank/wiki). For participants who answer a subsequent version of the questionnaire, these questions are related to their experience in the period since the previous questionnaire. Additional questions and question modules have been added as the questionnaire programme

progressed, including the Groningen Frailty Indicator,[8] KIDSCREEN-10[9] and the Positive and Negative Affects Schedule.[10]

Lifelines COVID-19 cohort data can be linked to other participant data held by Lifelines, including detailed biological measurements such as genotype, metagenomics, metabolomics and transcriptomics collected for subcohorts within Lifelines such as Lifelines DEEP[11] and Lifelines DAG3. Data can also be linked to the administrative records held by Statistics Netherlands (https://www.cbs.nl/en-gb), which include health-related records

**Table 1** Lifelines COVID-19 questionnaire

| Subject | Question | | Answer type |
|---|---|---|---|
| Sociodemographic | | | |
| Age | 1 | | In the Lifelines database |
| Sex | 2 | | In the Lifelines database |
| Location | 3. What are the four numbers of the postcode of your home address? | | Numerical field |
| Living situation | 4. The following questions are about your household members who live with you at least 1 day a week. | | |
| | 4a1. How many household members are between 0 and 12 years of age? | | Numerical field |
| | 4a2. How many household members are between 13 and 18 years of age? | | Numerical field |
| | 4a3. How many household members are between 18 and 30 years of age? | | Numerical field |
| | 4a3a. How many household members are between 19 and 30 years of age? | | Numerical field |
| | 4a4. How many household members are between 30 and 59 years of age? | | Numerical field |
| | 4a4a. How many household members are between 31 and 60 years of age? | | Numerical field |
| | 4a5. How many household members are older than 60 years of age? | | Numerical field |
| Effects children | if 4a1>0 or 4a2>0 | | |
| | 4b. Are your household members under 19 years of age your children or foster children? | | Yes/no/both |
| | if 4b = 'Yes' or 'Both' | | |
| | 4b1. Are your children (or child) experiencing stress about the corona crisis? | | No stress/some stress/much stress/a lot of stress |
| | if 4b1='some stress' or 'much stress' or 'a lot of stress' | | |
| | 4b1a. How do they show that? | | Text field |
| | if 4b= 'Yes' | | |
| | 4b2. Do your children (or child) feel safe at home? | | Safe/somewhat safe/unsafe/very unsafe |
| | 4b3. Do your children (or child) feel safe in their neighbourhood? | | Safe/somewhat safe/unsafe/very unsafe |
| Employment | 5. What is your current work situation? | | I am a student/I work (full time, part-time, freelance)/I am disabled/I am unemployed/I am retired |
| | if 5='I work' | | |
| | 5a. What kind of work contract do you have? | | Full time/part-time/zero hour, flexible, on call/freelance |
| | 5b. What is your current work situation? | | I am working from home/I am being paid to work from home/I have been laid off work without pay/I continue to work at the usual location (eg, office, factory, construction site)/I continue to work at multiple sites for my job/I have been forced to take sick leave or vacation time |
| | 5c. Do you have a critical job? (as defined by the government) | | Yes/no |
| | if 5b='I continue to work at the usual location' | | |
| | 5d. What is the location of your workplace? (postcode) | | Numerical field |

Continued

**Table 1** Continued

| Subject | Question | Answer type |
|---|---|---|
| | if 5='I am unemployed' | |
| | 5e. Are you unemployed because of the COVID-19 crisis? | Yes/no |
| | if 5='I work' | |
| | 5f. Since the start of the COVID-19 crisis in NL (mid-March), have you sometimes or regularly worked night shifts? | Yes, regularly/yes, sometimes/no |
| | 5g. Do you work in a profession in which you still come into frequent contact with patients, clients, children or the general public since the start of the COVID-19 crisis in the Netherlands (mid-March)? (For example, nursing, teaching, supermarket staff, police, emergency services and so on) | Yes/no |
| | 4.1 Do you have any other household members? *This applies to anyone who lives with you at least 1 day a week.* | Yes/no |
| | if 4a1 or 4a2 or 4a3 or 4a4 or 4a5 > 0 | |
| | 4c. Do any of your household members have a critical job? (As defined by the government.) | Yes/no |
| | 4d. Does at least one of your household members work outside the house? | Yes/no |
| Weight | 8a. What is your current weight (in kg)? If you have a scale in the house, please weigh yourself. | Numerical field, kg |
| | 8b. At what time of day did you weigh yourself? | morning/evening/I estimated my weight |
| Vaccinations | 9. Did you get an influenza shot in the past year? | Yes/no/don't know |
| | 10. Have you ever been vaccinated against tuberculosis? (BCG) | Yes/no/don't know |
| | if 11='Yes' | |
| | 10a. What year were you vaccinated against tuberculosis (give an estimate if not sure)? | Numerical field |
| **Chronic illnesses** | | |
| **Subject** | **Question** | **Answer type** |

Continued

**Table 1** Continued

| Illness | | Yes/no |
|---|---|---|
| | 1. Do you have a chronic health condition? | |
| | 1a. Cardiovascular disease (including high blood pressure) | |
| | 1a1. High blood pressure | |
| | 1a1. Heart attack | |
| | 1a1. Narrowing of the arteries in the legs | |
| | 1a1. Stroke or TIA | |
| | 1a1. Other heart and/or coronary disease | |
| | 1b. Lung disease, such as asthma, COPD or chronic bronchitis | |
| | 1c. Liver disease | |
| | 1d. Kidney disease or reduced kidney function | |
| | 1e. Diabetes | |
| | 1f. Chronic muscle disease | |
| | 1g. Psychological illness, such as depression, psychosis or anxiety disorder | |
| | 1h. Auto-immune illness, such as coeliac disease, inflammatory bowel disorder, rheumatoid arthritis, lupus | |
| | 1i. Cancer | |
| | 1j. Neurological disease, such as dementia, Parkinson's disease or Alzheimer's disease | |
| | 1k. Problems with your spleen (eg, sickle cell anaemia, spleen removed) | |
| | 1m. Do you have another kind of chronic condition? | |
| | 1m1. Specify other condition | Text |

| **COVID-19 related** | The following questions are about the period since the outbreak of the COVID-19. | |
|---|---|---|
| **Subject** | **Question** | **Answer type** |
| COVID-19 | 1a. Have you been tested for coronavirus (COVID-19)? | Yes/no |
| | if 1a="Yes" | |
| | 1a1. Do you have or have you had a coronavirus/COVID-19 infection? | Yes/no |
| | 1a1. What was the result of your corona virus (COVID-19) test? | Positive, I have a corona virus infection (COVID-19)/ negative, I do not have a corona virus infection (COVID-19) |
| | if 1a = 'No' | |
| | 1b. Has a doctor told you that you may have (or have had) a Covid-19 infection? | Yes/no |
| | 1c. Do you also think you have (or had) a Covid-19 infection? | Yes/no |
| | if 1a1='Yes' or 1b ='Yes' of 1 c ='Yes' | |
| | 1d. Do you know how you got the infection? | Household family member/other family member/friends/co-workers/sport/other/unknown |

Continued

**Table 1** Continued

| Subject | Question | Answer type |
|---|---|---|
| | 2a. Has someone you live with tested positive for a Covid-19 infection? | Yes/no |
| | 2b. Has someone you live with been told by a doctor that they might have Covid-19? | Yes/no |
| | 2c. Have you had contact with someone who tested positive for Covid-19? This means physical contact rather than by, for example, telephone. | Yes/not that I am aware of |
| | 2d. In the last 14 days have you had contact with someone who tested positive for Covid-19? This means physical contact rather than by, for example, telephone. | Yes/Yes, but I am a healthcare professional and used the appropriate personal protection equipment/Not that I am aware of |
| | 2e. Before filling in the previous questionnaire, had you had contact with someone who has been diagnosed COVID-19 in the interval between then and now? This person either had symptoms at the time of contact or in the previous 24 hours, or they were diagnosed within a week after contact. | Yes/Yes, but I am a healthcare professional and used the appropriate personal protection equipment/Not that I am aware of |
| Hospitalisation | if 1a1='Yes' or 1b='Yes' | |
| | 3. Have you been hospitalised for a Covid-19 infection? | Yes/no |
| | Have you been hospitalised for a Covid-19 infection since the last time you filled in the corona virus (COVID-19) questionnaire? | Yes/no |
| | if 3='Yes' | |
| | 3a. Were you given supplemental oxygen? | Yes/no |
| | 3b. Were you put on antibiotics? | Yes/no |
| | 3c1. Were you in the intensive care unit of the hospital? | Yes/no |
| | if 3c1='Yes' | |
| | 3c2. Were you put on a ventilator? | Yes/no |
| **Health** | | |
| **Subject** | **Question** | **Answer type** |
| Overall health | 1. How would you rate your health, in general? | excellent/very good/good/medium/poor |
| Recent symptoms | 2. To what degree have you experienced the following symptoms in the last 7 days: *(Please fill in these answers even if the symptoms are chronic for you or you think you had them for reasons other than a corona virus infection)* | |

Continued

| Table 1 | Continued | |
|---|---|---|
| | 2a. Headache | not at all/a little/some/quite a lot/often |
| | 2b. Dizziness | |
| | 2c. Heart or chest pain | |
| | 2d. Lower back pain | |
| | 2e. Nausea or upset stomach | |
| | 2f. Muscle pain/aches | |
| | 2g. Difficulty breathing | |
| | 2h. Feeling suddenly warm, then suddenly cold again | |
| | 2i. Numbness or tingling somewhere in your body | |
| | 2j. A lump in your throat | |
| | 2k. Part of your body feeling limp | |
| | 2l. A feeling of heaviness in your arms or legs | |
| | 2m. Shortness of breath | |
| | 2n. Pain when breathing | |
| | 2o. Runny nose | |
| | 2p. Sore throat | |
| | 2q. Dry cough | |
| | 2r. Wet cough | |
| | 2s. Fever (38 degrees or higher) | |
| | 2t. Diarrhoea or stomach pain | |
| | 2t1. Diarrhoea | |
| | 2t2. Stomach pain | |
| | 2u. Loss of sense of smell or taste | |
| | 2v. Red, painful or itchy eyes | |
| | 2w. Sneezing | |
| | 2x. Sensitive skin | |
| | 2y. Pain in neck, shoulder(s) or arm(s) | |
| | 2z. Upper back pain | |
| | 3. To what degree did you experience the following in the last 7 days: | |
| Fatigue | 3a. I felt tired | seven point NRS with left anchor "Yes, that's correct" and right anchor "No, that's not correct" |
| | 3b. I got tired quickly | |
| | 3c. I felt fine | |
| | 3d. I felt physically exhausted | |

Continued

**Table 1** Continued

| Subject | Question | Answer type |
|---|---|---|
| Sex | 4. Are you a woman between 18 and 55 years of age? *We do have this information in our database, but to ensure the rapid processing of this questionnaire, we are asking you to fill this in again.*<br><br>if 4=Yes | |
| Menstruation | 4a. Did you menstruate in the last 7 days? | Yes/no/Prefer not to say |
| Doctor avoidance | 5. In the last 7 days have you had health problems that you would normally see the doctor for, but chose not to contact your doctor? | Yes/no |
| | 6. What best describes these symptoms? | 1=Symptoms I had previously but haven't experienced in a while, 2=Intensification of existing symptoms, 3=New symptom(s), 4=New psychological condition, 5=Symptoms fit with a corona infection, 6=Other |
| | 7. Why did you choose not to contact your doctor? More than one answer is possible. | 1=Symptoms not bad enough, 2=I got information elsewhere, 3=I started self-treatment, 4=I was anxious about contracting corona, 5=I did not want to bother my doctor, 6=Financial concerns, 7=No time, 8=Other (More than one option is possible) |

**MEDICATION**

| Subject | Question | Answer type |
|---|---|---|
| | 10. Has your medication usage changed since the last time you filled in the corona questionnaire? Don't forget to think about over-the-counter medications like cough syrup or paracetamol. If you're not sure, click 'Yes'.<br><br>if 10=Yes | Yes/no |
| | Have you taken any medications in the last 7 days? | Yes/no |
| | Which medications have you taken in the last 7 days? | |
| | 1. High blood pressure medicine (such as metoprolol, furosemide, enalapril) | Yes/no |
| | 2. Inhaler | Yes/no |
| | 3. Corticosteroids in tablet form (such as prednisone) | Yes/no |
| | 4. Other corticosteroids (such as injections, hormone creams, eye or ear drops) | Yes/no |
| | 5. Cholesterol-lowering medication | Yes/no |
| | 6. Diabetes medication | Yes/no |
| | 7. Cough medicine | Yes/no |
| | 8. Pain medication | Yes/no |
| | 9. Other | Text |
| | if 1='Yes' | |

Continued

**Table 1** Continued

| | |
|---|---|
| Which blood pressuring lowering medications (eg, metoprolol, furosemide, enalapril) have you used in the last 7 days? *Multiple answers are possible.* | Hydrochlorothiazide, Furosemide (eg, Lasix), Bumetanide (eg, Burinex), Atenolol, Metoprolol (eg, Selokeen ZOC), Bisoprolol (eg, Emcor), Captopril, Enalapril (eg, Renitec), Lisinopril (eg, Zestril), Nifedipine |
| Other, specifically medicine 1: | Text |
| Other, specifically medicine 2: | Text |
| if 2='Yes' | |
| Which inhalers have you used in the last 7 days? *Multiple answers are possible.* | Salbutamol (eg, Ventolin, Airomir), Formoterol (eg, Oxis, Foradil), Salmeterol (eg, Serevent), Ipratropium (eg, Ipraxa, Atrovent), Tiotropium (eg, Spiriva), Beclometasone (eg, Qvar), Budesonide (eg, Pulmicort), Fluticasone (eg, Flixotide), Foster, Symbicort, Seretide |
| Other, specifically medicine 1: | Text |
| Other, specifically medicine 2: | Text |
| if 3='Yes' | |
| Which corticosteroids (such as prednisone) have you used in the last 7 days? *Multiple answers are possible.* | Cortisone, Dexamethasone, Hydrocortisone, Prednisolone, Prednisone |
| Other, specifically medicine 1: | Text |
| Other, specifically medicine 2: | Text |
| if 4='Yes' | |
| Which other corticosteroids (such injections, hormone creams or eye/eardrops) have you used in the last 7 days? *Multiple answers are possible.* | Injection with triamcinalonacetonide (eg, Kenacort-A), Salve or cream with triamcinolonacetonide, Nasal spray with triamcinolonacetonide (eg, Nasacort), Eardrops with triamcinolonacetonide, Salve or cream with hydrocortisone, Salve or cream fluticasone (eg, Cutivate), Salve or cream with betamethasone, Salve or cream with dexamethasone, Eyedrops with dexamethasone, TriAnal |
| Other, specifically medicine 1: | Text |
| Other, specifically medicine 2: | Text |
| if 5='Yes' | |
| Which cholesterol lowering medications have you used in the last 7 days? *Multiple answers are possible.* | Simvastatin (eg, Zocor), Atorvastatin (eg, Lipitor), Fluvastatin (eg, Lescol), Rosuvastatin (eg, Crestor), Pravastatin, Gemfibrozil (eg, Lopid), Cholestyramine (eg, Questran), Ezetimib (eg, Ezetrol), Inegy |
| Other, specifically medicine 1: | Text |
| Other, specifically medicine 2: | Text |
| if 6='Yes' | |

Continued

**Table 1** Continued

| | | |
|---|---|---|
| Which diabetes-related medications have you used in the last 7 days? *Multiple answers are possible.* | Insulin (eg, Novorapid, Novomix, Insulatard, Mixtard, Lantus), Metformin, Tolbutamide, Glibenclamide, Gliclazide (eg, Diamicron), Pioglitazone (eg, Actos), Repaglinide (eg, NovoNorm), Acarbose (eg, Glucobay), Sitagliptine (eg, Yesnuvia) | |
| Other, specifically medicine 1: | Text | |
| Other, specifically medicine 2: | Text | |
| if 7='Yes' | | |
| Which cough medicines have you used in the last 7 days? *Multiple answers are possible.* | Codeine, Noscapine, Broomhexine, Althea syrup or thyme syrup, Dextromethorphan, Pentoxyverine, Acetylcysteine, Carbocysteine, Promethazine, Chamomile or menthol | |
| Other, specifically medicine 1: | Text | |
| Other, specifically medicine 2: | Text | |
| if 8='Yes' | | |
| Which pain killers have you used in the last 7 days? *Multiple answers are possible.* | Paracetamol (acetaminophen), Ibuprofen (eg, Brufen), Acetylsalicylic acid (eg, Aspirin), Diclofenac, Naproxen (eg, Aleve), Codeine, Tramadol (eg, Tramal), Oxycodone (eg, OxyContin, OxyNorm), Morphine (eg, MS Contin, Oramorph) | |
| Other, specifically medicine 1: | Text | |
| Other, specifically medicine 2: | Text | |
| if 9='Yes' | | |
| 9a. How many other different medicines have you used in the last 7 days? *(maximum 10)* | Text | |

**Mental health and well-being**

| Subject | Question | Answer type |
|---|---|---|
| MINI – Depression | 1. In the last 7 days have you felt low or depressed for much of the day, every day? | Yes/no |
| | 2. In the last 7 days have you had the feeling that you've lost interest in or the will to do things you are normally interested in? | Yes/no |
| | 3. The following questions are about your experience in the last 7 days: | |
| | 3a. Did your appetite change noticeably, or did your weight increase or decrease without this being intended? | Yes/no |
| | 3b. Have you had problems sleeping almost every night (difficulty falling asleep, waking up in the night or too early in the morning, or actually sleeping too much)? | Yes/no |
| | 3c. Did you speak or move more slowly than normal? Or did you feel restless, jittery and could barely sit still? Nearly every day? Yes/no | |
| | 3d. Did you feel worthless or guilty almost every day? | Yes/no |

Continued

**Table 1** Continued

| | | Answer type |
|---|---|---|
| | 3e. Was it difficult to concentrate or make decisions almost every day? | Yes/no |
| | 3f. Have you considered hurting yourself, wished you were dead, or had suicidal thoughts? | Yes/no |
| MINI – Anxiety | 4. In the last 7 days, have you been worrying excessively and worrying about multiple problems of everyday life, at work, at home, in your immediate environment? | Yes/no |
| | if 4='Yes' | |
| | 4a. Were these worries present almost every day in the last 7 days? | Yes/no |
| | 4b. In the last 7 days did you find it hard to set these worries aside or did they prevent you from concentrating? | Yes/no |
| | 5. In the last 7 days did it often happen that… | |
| | 5a. You felt restless, jittery or nervous? | Yes/no |
| | 5b. You felt tense? | Yes/no |
| | 5c. You were particularly irritable? | Yes/no |

**Corona related well-being**

| Subject | Question | Answer type |
|---|---|---|
| Pandemic worries | 1. How much have you been concerned about the corona crisis in the past 7 days? | 1=not concerned, 10=extremely concerned |
| | 2a. I worry about getting sick myself | Never/almost never/sometimes/frequently/always or almost always |
| | 2b. I worry that someone close to me will get sick | |
| | 2c. I am concerned that I or my family will be in serious financial trouble | |
| | 2d. I worry that I will lose my job | |
| | 2e. I worry that it will be a long time before my life returns to normal | |
| | 2f. I am concerned that I can't see friends and family | |
| | 2g. I am worried for another reason | |
| | if 2g = 'sometimes/frequently/always or almost always' | |
| | 2g. For what other reason are you worried? | Text |
| Infection precautions | 3. What precautions are you taking to prevent the spread of the coronavirus? | Frequent hand washing/Use of hand disinfectant/Social distancing (except for household members)/Social distancing, including household members/Covering my mouth and nose in public/Avoiding public transport/Reduced travel/Other, specifically… |
| | if 3 = 'Other, specifically…' | |
| | 3_txt. What other precautions are you taking to prevent the spread of the coronavirus? | Text |
| Information sources | 4. Where have you been getting your information and advice from **in the last 7 days**? | Media (Newspaper, TV, radio)/Health authorities (eg, government, RIVM, WHO)/Social media (eg, Facebook, twitter, instagram)/Family and friends/Others |

Continued

**Table 1** Continued

| Subject | Question | Answer types |
|---|---|---|
| | if 4='Others, specifically…' | |
| | 4_txt. Where else have you been getting your information and advice from in the last 7 days? | Text |
| Perceptions | 5. Covid-19 threatens everyone in the Netherlands. | 1=totally disagree, 2=disagree slightly, 3=neutral, 4=agree slightly, 5=totally agree |
| | 6. Since the beginning of the Covid-19 crisis, I see others in my area, such as people in the neighbourhood or in shops, as a threat to my well-being. | |
| | 7. I have faith in the Dutch government's response to the corona crisis. | |
| Corona proximity | 8. Does someone close to you have a Covid-19 infection? | Yes/no |
| | 9. Has someone close to you died of a Covid-19 infection? | Yes/no |
| Quality of life | 10. How would you rate your quality of life over the last 7 days? | 1=terrible, 10=excellent |

**Social life**

| Subject | Question | Answer types |
|---|---|---|
| Social isolation | 1. How socially isolated have you felt in the last 7 days? | 1=not socially isolated, 10=extremely socially isolated |
| Loneliness | 2. Can you tell us about how you felt in the last 7 days? | |
| | 2a. How often did you feel excluded? | Almost never or never/ sometimes/ often |
| | 2b. How often did you feel isolated from others? | Almost never or never/ sometimes/ often |
| | 2c. How often did you feel alone? | Almost never or never/ sometimes/ often |

**Social relations**

| Subject | Question | Answer type |
|---|---|---|
| | Can you indicate how much you agreed with the statements below **in the last 7 days?** | |
| | 1. I feel connected to all Dutch people | 1=totally disagree, 2=disagree slightly, 3=neutral, 4=agree slightly, 5=totally agree |
| | 2. I feel connected to my neighbours, family and/or friends | |
| | 3. I get the help and support I need from my neighbours, family and/or friends | |
| | 4. I do everything I can to help others who are infected with Covid-19 | |
| | 5. I expect that others will do everything they can to help me if I get infected or ill with Covid-19 | |
| | 6. I do not feel obliged to comply with the government's corona measures | |
| | 7. I feel excluded by society | |
| | 8. I feel that I am not appreciated by others in society | |
| | 9. I am frustrated with how things are now going in society | 1=absolutely not, 7=very much |
| | 10. I am afraid that things will go wrong in our society | 1=absolutely not, 7=very much |

**Lifestyle changes**

| Subject | Question | Answer types |
|---|---|---|

Continued

**Table 1** Continued

| | Question | Answer type |
|---|---|---|
| **Eating patterns** | 1. How healthy are you eating compared with the period before the Covid-19 crisis? | 1=Much less healthy, 2=less healthy, 3=just as healthy, 4=healthier, 5=much healthier |
| | 2. How often do you eat per day? | 1=Less than 3 x per day, 2=3 x per day, 3=4 x per day, 4=5 x per day, 5=6 x per day, 6=7 x per day, 7=8 x per day, 8=more than 8 x per day |
| | 3. How important do you think healthy eating is compared with the period before the Covid-19 crisis? | 1=Much less important, 2=Less important, 3=Just as important, 4=More important, 5=Very important |
| **Exercise** | 4. Before the corona crisis, how many minutes of (relatively) intense activity did you do each week (eg, walking, biking or running)? | 1=less than 50 mins, 2=50–100 mins, 3=100–150 mins, 4=150–180 mins, 5=more than 180 min |
| | 5. In the last **7** days, how many minutes of (relatively) intense activity did you do (eg, walking, biking or running)? | 1=less than 50 mins, 2=50–100 mins, 3=100–150 mins, 4=150–180 mins, 5=more than 180 min |
| | 6. I do muscle and bone strengthening exercises, such as Nordic walking, jumping rope or weight training: | 1=More than in the period before the Covid-19 crisis, 2=Just as much as in the period before the Covid-19 crisis, 3=Less than in the period before the Covid-19 crisis |
| **Smoking** | 7. Have you smoked in the last 7 days? | Yes/no |
| **Alcohol** | 8. Have you drunk alcohol in the last 7 days? If yes, how many units, on average? | 0=0, 1=1 glass or 'once', 2=2 glasses, 3=3 glasses, 4=4 glasses, 5=5 glasses, 6=6 or more glasses |
| | 8a. How many units of alcohol have you consumed in the last 7 days? | 0=0, 1=1 glass or 'once', 2=2 glasses, 3=3 glasses, 4=4 glasses, 5=5 glasses, 6=6 or more glasses |
| | 8b. How many units of alcohol have you consumed in the last 7 days total? | Fill in number, greater than 0, warning by 50 |
| **Activity levels** | 9. Before the corona crisis, how much time did you spend sitting, on average, per working day (Monday to Friday)? | Don't know, less than 1 hour, 1 hour, …, 12 hours, more than 12 hours |
| | 10. Before the corona crisis, how much time did you spend sitting, on average, per weekend day (Sat, Sun)? | Don't know, less than 1 hour, 1 hour, …, 12 hours, more than 12 hours |
| | 11. In the past 7 days, how much time did you spend sitting, on average, per working day (Monday to Friday)? | Don't know, less than 1 hour, 1 hour, …, 12 hours, more than 12 hours |
| | 12. In the past 7 days, how much time did you spend sitting on average per weekend day (Sat, Sun)? | Don't know, less than 1 hour, 1 hour, …, 12 hours, more than 12 hours |

**Comments**

| Question | Answer type |
|---|---|
| 1. Do you have any comments regarding this questionnaire? | Yes/no |
| if 1=Yes | |
| 1_txt. What comments do you have about this questionnaire? | Text |

**Groningen frailty index** – For respondents 65 years of age and older.

| Question | Answer type |
|---|---|
| 1. Are you 65 or older? We do have this information in our database, but to ensure the rapid processing of this questionnaire, we are asking you to fill this in again. | Yes/no |

Continued

**Table 1** Continued

| Question | Answer type |
|---|---|
| 2. Can you independently perform the following activities without any help from someone else, possibly with the help of a cane, walker or wheelchair? | Yes/no |
| 2a. Get groceries and run errands | Yes/no |
| 2b. Get dressed/undressed | Yes/no |
| 2c. Move outdoors (around house, to neighbours) | Yes/no |
| 2d. Go the toilet | Yes/no |
| 3. What score would give your fitness (from 0 to 10)? | Scale from 0 to 10 |
| 4. Do you have problems in everyday life due to poor vision? | Yes, many problems/Yes, some problems/No, no problems |
| 5. Do you have problems in everyday life due to poor hearing? | Yes, many problems/Yes, some problems/No, no problems |
| 6. Have you lost a lot of weight in the past period without wanting to (6 kg in 6 months or 3 kg in 1 month)? | Yes/no |
| 7. Do you have memory problems? | No/Sometimes/Yes |

**Kidscreen**
If household members are <18 years old (questions from the socio-demographic module).

| Question | Answer type |
|---|---|
| How old is your oldest child aged 18 years or younger? | Number |
| Complete the following questions for this child for the last 7 days: | |
| 1. Has your child felt fit and healthy? | Never/almost never/sometimes/frequently/always |
| 2. Has your child felt full of energy? | |
| 3. Has your child felt sad? | |
| 4. Has your child felt lonely? | |
| 5. Has your child had sufficient time for himself or herself? | |
| 6. Has your child been able to do the things her or she wanted to do in their free time? | |
| 7. Has your child felt that he or she has been treated fairly by his or her parents? | |
| 8. Did your child have fun with his or her friends? | |
| 9. Did school activities go well? | |
| 10a. Has your child been able to pay attention? | |
| 10b. Has your child felt anxious? | |
| 10c. Has your child felt angry? | |
| 10d. Has your child been bored? | |
| 11. In general, how would your child rate his or her health? | Excellent/very good/good/fair/poor |

Questions asked are modified slightly when respondents have filled in a previous questionnaire to indicate that they should answer with respect to the intervening period.
COPD, chronic obstructive pulmonary disease; MINI, Mini International Neuropsychiatric Interview; NL, The Netherlands; TIA, transient ischaemic attack.

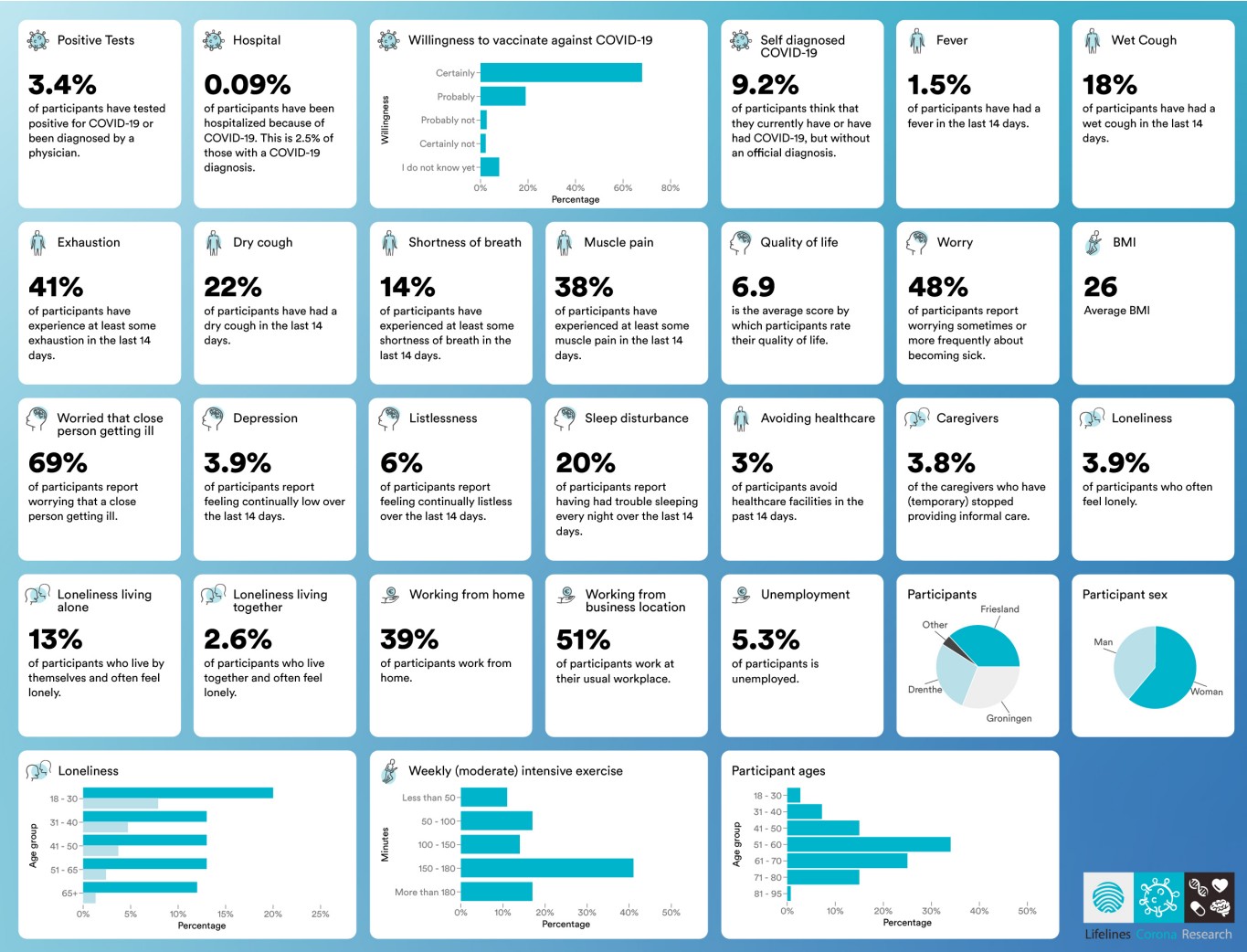

**Figure 4** Communicating COVID-19 cohort results to the public through the Corona Barometer. Snapshot of the Corona Barometer (https://coronabarometer.nl/), which is updated after every questionnaire round to present the most recent findings of the lifelines COVID-19 questionnaire in a format accessible by the public. The website is now interactive to enable users to look at trends over time and compare variables.

on mortality, hospital admissions and healthcare costs, as well as data on employment status, income, wealth and other sociodemographic characteristics. Data can also be linked to drug prescription data held by IADB.nl via the Pharmlines initiative[12] and to SARS-CoV-2 testing data (including serological data) held by Certe and other Dutch laboratories.

## Participant and public involvement

Projects carried out within Lifelines are discussed with the Lifelines Participant Advisory Board. With respect to the COVID-19 cohort, compiled questionnaire results have been continuously updated after each question-naire round and shared with participants and the public through interactive infographics on the Corona Barom-eter website (https://coronabarometer.nl/, snapshot in figure 4), as well as via frequent social media posts, press releases and interviews in the national press. Subsequent questionnaires have also been modified and refined in response to questions from participants. In addition, new

questionnaire modules have been added based on the results of previous rounds and to examine the effects of changes in national policy.

## Findings to date
### Response rates and characteristics of respondents
In every questionnaire round, 139 679 out of 159 482 current adult Lifelines participants are invited to respond to the COVID-19 questionnaire. In total, 71 992 (51.4%) Lifelines and LLNEXT participants responded to at least one of the first seven questionnaires, and response rates ranged from 33% to 39% (42 917–54 525) (figure 3). Compared with non-invited subjects (those without a known email address), invited subjects are younger, slightly more often female, have a lower body mass index (BMI) and are more often never smokers (table 2). Of the 139 679 Lifelines participants invited, 71 833 (51%) completed at least one of the questionnaires in the first 8 weeks of the programme. Compared with non-responders, these responders were slightly older, slightly

**Table 2** Characteristics of adult Lifelines participants invited to participate in the cohort and the participants of the COVID-19 questionnaire cohort during the first 8 weeks of the project (questionnaire rounds 1–7)

| | Invited | Not invited | OR (95% CI)* | P value | Responded | Not responded | OR (95% CI)* | P value |
|---|---|---|---|---|---|---|---|---|
| N (%) | 139 679 (87.6) | 19 803 (12.4) | | | 71 833 (51.4) | 67 846 (48.6) | | |
| Current age, mean (SD) | 51.1 (13.6) | 57.8 (18.3) | | <0.01† | 54.1 (12.9) | 47.9 (13.6) | | <0.01† |
| Male sex, % | 41.6 | 42.3 | 0.97 (0.95 to 1.00) | 0.08* | 39.2 | 44.2 | 0.81 (0.80 to 0.83) | <0.01* |
| BMI at last visit, mean (SD) | 25.9 (4.3) | 26.5 (4.9) | | <0.01† | 26.0 (4.3) | 25.8 (4.4) | | <0.01† |
| Smoking at last visit, % | | | | | | | | |
| Never | 52.1 | 46.1 | reference | | 51.8 | 52.6 | reference | |
| Ex | 32.1 | 37.4 | 0.76 (0.71 to 0.81) | <0.01* | 34.0 | 28.6 | 1.21 (1.17 to 1.25) | <0.01* |
| Current | 15.8 | 16.5 | 0.85 (0.78 to 0.93) | <0.01* | 14.2 | 18.8 | 0.76 (0.73 to 0.80) | <0.01* |

*OR, 95% CI and p value using a logistic regression analysis.
†P value using an independent t-test.
BMI, body mass index.

more often female, had a higher BMI, were less often current smokers and more often ex-smokers (table 2). While Lifelines as a whole has been shown to be representative of the regional population,[7] these slight differences in cohort makeup should be considered when looking at data from the COVID-19 cohort.

For LLNEXT, 321 people were invited to participate in the first 8 weeks of the project, and 159 participated (49.5%) (table 3). Compared with invitees, respondents were more likely to be female (73.5% of respondents vs 50.5% of invitees) (table 3). As LLNEXT recruits women who are currently pregnant, the age range was small and did not differ substantially between invitees, respondents and non-respondents. In LLNEXT, 80% of all parents who responded to the main Lifelines COVID-19 questionnaire also returned data on their children for the LLNEXT-specific module on COVID-19 and its impact on children. In total, we have data for 112 children up until week 8 of the COVID-19 questionnaire initiative: 96 children 0–3 years of age, 14 children 4–7 years of age and 2 children in the 8–18 age group.

### COVID-19 cases

Of the participants who responded in the first 8 weeks of the project, 1294 (1.8%) responded that they had been tested for COVID-19 and 127 (0.2%) tested positive. In addition, 887 (1.3%) respondents said they had been told by a doctor that they probably had COVID-19, while 5271 (7.3%) participants responded that they thought they had had COVID-19.

### Early results and ongoing projects

One of the earliest results of the project was a clear signal that feelings of loneliness and isolation were substantially stronger in individuals who lived alone and that this effect was strongest in the youngest age group of respondents (18–30 years old, see corresponding panel in figure 4). There was also an increase in the number of unemployed respondents who reported losing their jobs due to the crisis, rising from 7.5% of unemployed respondents to the first questionnaire round (Q1) up to 14% by the week 5 questionnaire (Q5). More recent results have shown that by the end of May 2020, as the number of infections and hospitalisations dropped to low levels and schools and business reopened, respondents were reporting less anxiety, better sleep and fewer worries about losing their jobs.

The data collected from the Lifelines COVID-19 cohort is currently being analysed to address the four goals of the project. COVID-19 cases reported by participants are being used to track the outbreak, and the symptoms reported by participants are being used to generate a symptom-based COVID-19 prediction model.[13] The data on chronic diseases, medication use and environmental factors (eg, cohabitation or smoking) will be used to look for associations with SARS-CoV-2 susceptibility and COVID-19 severity to help identify risk factors, protective factors and comorbidities. While factors such as age, sex, BMI and

**Table 3** Characteristics of LLNEXT participants invited to participate in the cohort and the participants of the COVID-19 questionnaire cohort during the first 8 weeks of the project (questionnaire rounds 1–7)

| | Invited | Responded | Not responded | Or (95% CI)* | P value |
|---|---|---|---|---|---|
| N (%) | 321 (100) | 159 (49.5) | 162 (50.5) | | |
| Current age, mean (SD) | 33.6 (4.9) | 33.0 (4.3) | 34.3 (5.3) | | 0.01† |
| Male sex, % | 49.5 | 36.5 | 62.3 | 0.35 (0.22 to 0.55) | <0.01* |

*OR, 95% CI and p value using a logistic regression analysis.
†P value using an independent t-test.

certain chronic illnesses have been associated with a more severe COVID-19 and higher mortality,[14] there have also been questions about whether recent vaccinations can be protective,[15 16] and cohort data should help address this. Important questions about why pregnant women and children seem to be relatively protected will also be analysed. Finally, it will be possible to look at genetic factors in detail as ~18 000 of the 71 922 COVID-19 participants who completed at least one questionnaire have been genotyped through the UMCG Genetics Lifelines Initiative. Next steps include identifying participants who were also participants in the Lifelines cohorts for which we have more detailed data, for example, participants with gut microbiome data, currently available for >10 000 Lifelines participants.

Mental health problems are known to increase in times of physical and psychological distress. The current COVID-19 pandemic is accompanied by strict government measures of social distancing and quarantine. As these events place significant stress on society and increase isolation and loneliness, close monitoring of mental well-being is important for both short-term and long-term public health policies and individual-level care. Alertness in clinical systems and tailored mental healthcare may be needed during and after such a mass traumatic event. The data from the Mini International Neuropsychiatric Interview major depressive disorder (MD) and general anxiety disorder (GAD)[17] modules and the societal impact modules of the questionnaire will allow researchers to: (1) longitudinally track the prevalence of symptoms and diagnoses of MD and GAD during the pandemic in the Lifelines and LLNEXT populations, (2) associate symptoms with COVID-19 severity and outcome, (3) identify at-risk groups and individuals and (4) measure the impact of government policies on the overall mental health in the cohort.

The questionnaire will also help address the major impact the pandemic has had on the working lives of people in the Netherlands. Healthcare workers are a particularly vulnerable group due to their higher risk of being infected by SARS-CoV-2 and their working conditions during the height of infections included long work hours, cancelled holidays, adverse physical and psychosocial work conditions, that is, high psychological and emotional demands and low control.[18 19] These working conditions, together with moral distress in relation to the family situation, may increase the risk for mental

health problems and sickness absence in this female-dominated occupational group with a high baseline risk. Other 'essential' occupational groups are experiencing unprecedented changes in their working environments that may affect their physical and mental health as well as their labour market attachment. For many 'non-essential' occupational groups that are now encouraged to work from home, the home working environment might not be suitable, and many families now have to combine working from home with caring for children. This will likely impact the productivity and quality of their work and their level of stress.

The lockdown has led to a sudden disruption of the economy, with several economic sectors effectively brought to a standstill. The Lifelines COVID-19 questionnaire is monitoring changes in people's current work situation by asking if they lost their job because of the crisis, if they are working in an essential job and whether they have to work from home. The answers to these questions will be used to monitor both the impact of the crisis on the short-term and longer term labour market and to identify workers most at risk of losing their job. This is essential information for policymakers to be able to target measures to the most vulnerable groups in society and mitigate the financial impact of the crisis.

### Strengths and limitations

One of the main strengths of this cohort is its embedding within the long-running Lifelines prospective population cohort, which provides a rich data background about participants and the knowledge, infrastructure and relationship with participants necessary to recruit and engage participants during an evolving crisis. The high and sustained rate of response and the weekly questionnaires mean that the cohort can provide a detailed longitudinal prospective view of both the outbreak and the long-term impacts of the crisis. Another strength is the collaboration of researchers across a range of disciplines in designing and implementing a questionnaire that can be used to address a wide range of research questions, can have immediate impact on policy and can be used to help design new policies to prevent and/or manage renewed outbreaks. Finally, Lifelines will continue to follow its participants for the coming decade and beyond, providing opportunities to examine the long-term health impacts of SARS-CoV-2 infection and of the pandemic.

The Lifelines COVID-19 questionnaire was also designed to make comparisons with similar projects throughout Europe. Direct cross-national comparisons with projects in Denmark and France are possible, as they are using nearly identical questionnaires and will provide unique opportunities to examine the effects of different governmental measures on mental health and well-being. The cohort is part of COVID-MINDS network of longitudinal studies on the global mental health impact of COVID-19 (https://www.covidminds.org/),[20] and the Lifelines COVID-19 project is also participating in the COVID-19 Host Genetics Initiative,[21] an international collaboration to share and analyse data to identify the genetic determinants of SARS-CoV-2 susceptibility, COVID-19 severity and outcomes. In addition, the Lifelines COVID-19 questionnaires have been requested by other (inter)national researchers as a basis for designing their own questionnaires, for example, separate research has been done on the experiences of COVID-19 patients, making use of the Lifelines COVID-19 questions.

The timing and nature of the COVID-19 outbreak in the Northern Netherlands, which diverged from that in other parts of the country, is both a strength and a limitation. The relatively low number of cases in the region, even accounting for undiagnosed cases, may seem to pose difficulties for statistical association analyses looking at COVID-19 related factors. However, as of June 2020, >800 participants reported having had COVID-19, as confirmed by a positive test or a doctor's diagnosis, which permits a wide number of statistical association analyses. Moreover, the impact of the societal steps taken to reduce the rate of infection in more heavily impacted regions of the Netherlands and the impact of the associated economic crises should have similar psychological and social impacts in the Lifelines population. The fact that the initial outbreak in the north was effectively capped by public health steps now puts the questionnaire programme in an interesting position to monitor the immediate health and societal impacts of the lockdown measures and the impact of coming out of lockdown. It also lays the groundwork for steps to be taken if there is a resurgence of COVID-19 infections, and the data generated while local infection rates were low could work as baseline values if subsequent outbreaks in the northern provinces are more intense.

**Author affiliations**
[1]Department of Genetics, University of Groningen, University Medical Center Groningen, Groningen, The Netherlands
[2]Department of Genetics, University Medical Centre Utrecht, Utrecht, The Netherlands
[3]Department of Epidemiology, University of Groningen, University Medical Center Groningen, Groningen, The Netherlands
[4]Department of Psychiatry, University of Groningen, University Medical Center Groningen, Groningen, The Netherlands
[5]Department of Cardiology, University of Groningen, University Medical Center Groningen, Groningen, The Netherlands
[6]Department of Health Sciences, Community and Occupational Medicine, University of Groningen, University Medical Center Groningen, Groningen, The Netherlands
[7]Faculty of Economics and Business, University of Groningen, Groningen, The Netherlands
[8]Department of Nephrology, University Medical Center Groningen, Groningen, The Netherlands
[9]Lifelines Cohort Study, Groningen, The Netherlands
[10]Department of Obstetrics and Gynaecology, University of Groningen, University Medical Center Groningen, Groningen, The Netherlands
[11]K.G. Jebsen Coeliac Disease Research Centre, Institute of Clinical Medicine, University of Oslo, Oslo, Norway
[12]Center of Development and Innovation, University of Groningen, University Medical Center Groningen, Groningen, The Netherlands
[13]Aletta Jacobs School of Public Health, University of Groningen, Groningen, The Netherlands

**Acknowledgements** We would like to thank all Lifelines and Lifelines NEXT participants for repeatedly filling out our questionnaires and all the experts involved in developing the content of the questionnaires. We would like to thank the Applied Health Research unit of the UMCG Department of Health Sciences, the UMCG Genomics Coordination Center, the UG Center for Information Technology and their sponsors BBMRI-NL & TarGet for storage and computing infrastructure. We would also like to thank Alex Friedrich and Gerolf de Boer for critical input for figure 1 and supplying the R(t) data in the figure and Trishla Sinha for editing the Lifelines NEXT children's COVID-19 questionnaires.

**Contributors** CW, JAMD, JM, HMB and LF conceived and implemented the study. KMI, PL, PD, HHW, JMV, APSO, SAJ, RW, IvB, FB, MXLD, JCH, AC, OB, EALM, UB, AZ, SAR, EZ, MAS, SB, RvO, VA, LHD, AS, SAS, JAMD, JM, HMB and LF contributed to the design and content of the questionnaire. PL, PD, HHW, JMV, APSO, SAJ and RW carried out the data analyses, established the Corona Barometer website and provided all figures and tables. KMI drafted the manuscript with contributions from PL, PD, HHW, JMV, APSO, SAJ, RW, IvB, FB, MXLD, JCH, AC, OB, EALM, UB, AZ, SAR, EZ, MAS, SB, RvO, VA, LHD, AS, SAS, JAMD, JM, HMB and LF. All authors reviewed and edited the manuscript and approved the final version.

**Funding** The Lifelines COVID-19 cohort is self-funded with in cash and in kind contributions from the initiators. The LLNEXT cohort is funded with a contribution of the UMCG Fund hereditary metabolic disorders (RVB16.0120). LF is supported by a Netherlands Organisation for Scientific Research (NWO) Corona Fast-Track grant (440.20.001), an Oncode Senior Investigator grant, a grant from the European Research Counsel (ERC Starting Grant agreement number 637640 ImmRisk) and an NWO VIDI grant (917.14.374). AZ is supported by the ERC Starting Grant 715772, NWO-VIDI grant 016.178.056, the Netherlands Heart Foundation CVON grant 2018-27 and the NWO Gravitation grant ExposomeNL 024.004.017. JM and LHD are supported by an NWO Fast-track grant (440.20.002). MAS is supported by the EU Horizon2020 EUCAN-connect programme (824989).

**Competing interests** None declared.

**Patient consent for publication** Not required.

**Provenance and peer review** Not commissioned; externally peer reviewed.

**Data availability statement** Data are available on reasonable request. The data analysed in this study were obtained from the Lifelines biobank, under project application number ov20_0554. Researchers interested in using these data should contact the Lifelines Research Office (research@lifelines.nl).

**ORCID iDs**
Katherine Mc Intyre http://orcid.org/0000-0001-6627-2920
Pauline Lanting http://orcid.org/0000-0002-7288-4445
Patrick Deelen http://orcid.org/0000-0002-5654-3966
Henry H Wiersma http://orcid.org/0000-0001-8353-1197
Judith M Vonk http://orcid.org/0000-0001-7531-4547
Anil P S Ori http://orcid.org/0000-0003-0579-0905
Soesma A Jankipersadsing http://orcid.org/0000-0001-9225-9236
Robert Warmerdam http://orcid.org/0000-0001-8691-0053
Irene van Blokland http://orcid.org/0000-0002-8545-8015
Floranne Boulogne http://orcid.org/0000-0002-5893-1401
Johanna C Herkert http://orcid.org/0000-0003-0461-910
Annique Claringbould http://orcid.org/0000-0002-9201-6557
Olivier Bakker http://orcid.org/0000-0002-1447-1327
Esteban A Lopera Maya http://orcid.org/0000-0001-5862-3938
Ute Bültmann http://orcid.org/0000-0001-9589-9220
Alexandra Zhernakova http://orcid.org/0000-0002-4574-0841
Sijmen A Reijneveld http://orcid.org/0000-0002-1206-7523
Elianne Zijlstra http://orcid.org/0000-0002-9845-614X
Morris A Swertz http://orcid.org/0000-0002-0979-3401
Raun van Ooijen http://orcid.org/0000-0002-0654-5568
Viola Angelini http://orcid.org/0000-0002-5177-3447
Sicco A Scherjon http://orcid.org/0000-0002-6902-1235
Cisca Wijmenga http://orcid.org/0000-0002-5635-1614
Jochen Mierau http://orcid.org/0000-0002-9968-8807
H Marike Boezen http://orcid.org/0000-0002-4320-1481
Lude Franke http://orcid.org/0000-0002-5159-8802

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
