## [Reviewer comments · BMJ Open]

ARTICLE DETAILS

TITLE (PROVISIONAL)	Cohort Profile: The Lifelines COVID-19 Cohort: investigating COVID-19 infection and its health and societal impacts in a Dutch population-based cohort
AUTHORS	Mc Intyre, Katherine; Lanting, Pauline; Deelen, Patrick; Wiersma, Henry; Vonk, Judith; Ori, Anil; Jankipersadsing, Soesma; Warmerdam, Robert; van Blokland, Irene; Boulogne, Floranne; Dijkema, Marjolein; Herkert, Johanna; Claringbould, Annique; Bakker, Olivier; Lopera Maya, Esteban; Bultmann, Ute; Zhernakova, Alexandra; Reijneveld, Sijmen; Zijlstra, Elianne; Swertz, Morris; Brouwer, Sandra; van Ooijen, Raun; Angelini, Viola; Dekker, Louise; Sijtsma, Anna; Scherjon, Sicco; Wijmenga, Cisca; Dekens, Jackie; Mierau, Jochen; Boezen, H. Marike; Franke, Lude

VERSION 1 – REVIEW

REVIEWER	ASSOC. PROF. STELIOS ZIMERAS University of Aegean Dept. of Statistics and Actuarial-Financial Mathematics GREECE
REVIEW RETURNED	07-Nov-2020

GENERAL COMMENTS	The statistical analysis is not very clearly. For the Table 2 and 3 results the authors they have not analysed the methodology, Also risk measures like OR could be used for smokers or BMI. Final a model between them could be useful
---

REVIEWER	Esther Garcia Garcia-Esquinas Spain
REVIEW RETURNED	17-Nov-2020

GENERAL COMMENTS	This article describes un detail the profile of the Lifelines COVID-19 Cohort in The Netherlands. The paper is well written and the net
---

VERSION 1 – AUTHOR RESPONSE

Reviewer: 1

Dr. Stelios Zimeras, Panepistemio Aigaiou

Comments to the Author:

The statistical analysis is not very clearly. For the Table 2 and 3 results the authors they have not analysed the methodology, Also risk measures like OR could be used for smokers or BMI. Final a model between them could be useful

We thank Dr. Zimeras for his suggestions and comments. We have now added Odds Ratios to both Tables 2 and 3 for the categorical variables (male sex and smoking). We have also included the methodology used (independent T-test for continuous variables and logistic regression models for categorical variables) as a footnote to the tables.

Reviewer: 2

Dr. E Garcia-Esquinas, Carlos III Health Institute

Comments to the Author: This article describes in detail the profile of the Lifelines COVID-19 Cohort in The Netherlands. The paper is well written and the net [truncated]

We appreciate Dr. Garcia-Esquinas's comment that the paper is well written. As the truncated comments were not available, we hope that our changes in response to Reviewer 1's comments cover any concerns that Dr. Garcia-Esquinas might have had.